# The Effect of Metal Cations on the Aqueous Behavior of Dopamine. Thermodynamic Investigation of the Binary and Ternary Interactions with Cd^2+^, Cu^2+^ and UO_2_^2+^ in NaCl at Different Ionic Strengths and Temperatures

**DOI:** 10.3390/molecules26247679

**Published:** 2021-12-19

**Authors:** Antonio Gigliuto, Rosalia Maria Cigala, Anna Irto, Maria Rosa Felice, Alberto Pettignano, Concetta De Stefano, Francesco Crea

**Affiliations:** 1Dipartimento di Scienze Chimiche, Biologiche, Farmaceutiche ed Ambientali, Università degli Studi di Messina, V.le F. Stagno d’Alcontres, 31, I-98166 Messina, Italy; agigliuto@unime.it (A.G.); rmcigala@unime.it (R.M.C.); airto@unime.it (A.I.); mrfelice@unime.it (M.R.F.); cdestefano@unime.it (C.D.S.); 2Dipartimento di Fisica e Chimica, Università degli Studi di Palermo, V.le delle Scienze, ed. 17, I-90128 Palermo, Italy; alberto.pettignano@unipa.it

**Keywords:** chemical speciation, metal complexes, catechol, sequestration, stability constants

## Abstract

The interactions of dopamine [2-(3,4-Dihydroxyphenyl)ethylamine, (Dop^−^)] with cadmium(II), copper(II) and uranyl(VI) were studied in NaCl(aq) at different ionic strengths (0 ≤ *I*/mol dm^−3^ ≤ 1.0) and temperatures (288.15 ≤ *T*/K ≤ 318.15). From the elaboration of the experimental data, it was found that the speciation models are featured by species of different stoichiometry and stability. In particular for cadmium, the formation of only MLH, ML and ML_2_ (M = Cd^2+^; L = dopamine) species was obtained. For uranyl(VI) (UO_2_^2+^), the speciation scheme is influenced by the use of UO_2_(acetate)_2_ salt as a chemical; in this case, the formation of ML_2_, MLOH and the ternary MLAc (Ac = acetate) species in a wide pH range was observed. The most complex speciation model was obtained for the interaction of Cu^2+^ with dopamine; in this case we observed the formation of the following species: ML_2_, M_2_L, M_2_L_2_, M_2_L_2_(OH)_2_, M_2_LOH and ML_2_OH. These speciation models were determined at each ionic strength and temperature investigated. As a further contribution to this kind of investigation, the ternary interactions of dopamine with UO_2_^2+^/Cd^2+^ and UO_2_^2+^/Cu^2+^ were investigated at *I* = 0.15 mol dm^−3^ and *T* = 298.15K. These systems have different speciation models, with the MM’L and M_2_M’L_2_OH [M = UO_2_^2+^; M’ = Cd^2+^ or Cu^2+^, L = dopamine] common species; the species of the mixed Cd^2+^ containing system have a higher stability with respect the Cu^2+^ containing one. The dependence on the ionic strength of complex formation constants was modelled by using both an extended Debye–Hückel equation that included the Van’t Hoff term for the calculation of the formation enthalpy change values and the Specific Ion Interaction Theory (SIT). The results highlighted that, in general, the entropy is the driving force of the process. The quantification of the effective sequestering ability of dopamine towards the studied cations was evaluated by using a Boltzmann-type equation and the calculation of pL_0.5_ parameter. The sequestering ability was quantified at different ionic strengths, temperatures and pHs, and this resulted, in general, that the pL_0.5_ trend was always: UO_2_^2+^ > Cu^2+^ > Cd^2+^.

## 1. Introduction

Dopamine (Figure 1) is a therapeutic form of a substance that naturally occurs in the body. It works by improving the pump force of the heart and improves blood flow to the kidneys. Dopamine is used to treat certain conditions that occur when there is a shock, which can be caused by heart attack, trauma, surgery, heart failure, kidney failure, and other serious medical conditions [1,2,3,4]. Dopamine is an important neurotransmitter of the catecholamine family, with a control function on movement, the so-called working memory, the sensation of pleasure, the production of prolactin, sleep regulation mechanisms, some cognitive functions, and the ability of attention. It also allows the cells of the nervous system to communicate with each other.

In the human body, the production of dopamine is mainly due to the so-called neurons of the dopaminergic area and, to a lesser extent, to the medullary portion of the adrenal glands.

The dopaminergic area includes several sites of the brain, including the pars compacta of the substantia nigra and the ventral tegmental area of the midbrain.

Abnormal dopamine levels are responsible for several pathological conditions, such as Parkinson’s disease [1,2,3,4,5].

Dopamine is also the precursor molecule from which cells, by means of specific processes, two other neurotransmitters of the catecholamine family derive: norepinephrine (or noradrenaline) and epinephrine (or adrenaline).

In neurons, neurotransmitters reside inside small vesicles; the vesicles are comparable to sacs, delimited by a double layer of phospholipids, very similar to that of the cytoplasmic membrane of a generic healthy eukaryotic cell.

Inside the vesicles, the neurotransmitters remain as it were inert, until a nerve impulse arrives in the neurons in which they reside.

The nerve impulses, in fact, stimulate the release of the vesicles by the neurons that contain them.

With the release of the vesicles, the neurotransmitters escape from the nerve cells, occupy the so-called synaptic space, and interact with neighboring neurons, to be precise with the membrane receptors of the aforementioned neurons. The interaction of neurotransmitters with neurons placed in the immediate vicinity transforms the initial nerve impulse into a very specific cellular response, which depends on the type of neurotransmitter and of receptors present on the neurons involved.

For this reason, neurotransmitters are chemical messengers, who nerve impulses released to induce a certain cellular mechanism.

The behavior in aqueous solutions, including biological fluids, of these organic molecules is strictly dependent on the chemical form in which they are present and on the possible presence of cationic species [6,7,8]. Therefore, to have information on their bioavailability, transport and eventual toxicity, it is necessary to know their speciation in those experimental conditions.

As it is known, the presence of many metals in biological fluids can have a double effect; the first one is to perform their function as essential metals for the various metabolisms that occur in our organism. The second effect is related to their concentration in the mentioned fluids. In fact, if they are present in high concentrations, a series of chronic and acute syndromes can be caused. However, even when non-essential or toxic metals are present in biological fluids in concentrations that do not directly cause damage to our body, they can still have a harmful effect deriving from their interactions with molecules of biological importance, such as drugs, rather than proteins or neurotransmitters, inhibiting their functions. For this reason, some years ago our research group has undertaken a systematic study on the protonation, solubility and interactions of biologically or pharmacologically relevant molecules with metals and just some of the results are reported in Refs. [9,10,11,12,13]. Furthermore, to modulate the molecules affinity towards cells, biological membranes and increasing their metal chelating affinity with respect to commercially available products, some new compounds have been designed and synthesized. Then, their behavior in aqueous solution and in the presence of metals has been investigated and tests in vivo performed to verify their actual efficacy [14,15,16,17,18].

This work can be considered as the continuation of an investigation already started on the ability of dopamine to interact with metals [12] in aqueous solution containing sodium chloride at different ionic strength and temperature.

The results obtained from those studies allowed us to model the dependence of the formation constants on the ionic strength and temperature by means of different mathematical approaches, allowing the determination of the enthalpies and entropies of formation of the species. The main results can be summarized: 1. as the cation varies, different speciation models have been obtained with differences of some orders of magnitude in terms of stability of the complexes; 2. in the case of methylmercury(II), calcium and magnesium, the speciation models contain mononuclear species; 3. for Sn^2+^, the speciation model is also formed by binuclear species and by ternary hydrolytic species; 4. the ions of the ionic medium have an influence on the distribution and stability of the species, especially for methylmercury(II) and tin(II); 5. a clear difference in the sequestering ability, calculated by means of the parameter pL_0.5_, was observed between the different metal-dopamine systems and with varying the pH, temperature and ionic strength.

Considering the importance of the results obtained from the previous investigation [12] and from similar studies carried out on adrenaline [11], it was very important to verify the behavior of dopamine when in the presence of cations such as copper(II), cadmium(II), or uranyl(VI).

Furthermore, since biological fluids are multicomponent solutions in which there is the simultaneous presence of metal cations, another objective of our study was to verify what could happen when two cations were simultaneously present in aqueous solution. However, these investigations on ternary systems of MM’L (M = UO_2_^2+^, M’ = Cu^2+^ or Cd^2+^, L = dopamine) type were carried out only at an ionic strength value (*I* = 0.15 mol dm^−3^) and temperature *T* = 298.15 K.

## 2. Results

### 2.1. Acid-Base Properties of Dopamine and Hydrolytic Constants of the Investigated Cations

In a previous paper [19], the protonation constants and the distribution of dopamine, between two different solvents, were already determined at different ionic strengths and temperatures.

The hydrolytic constants of Cd^2+^ and the formation constants of the CdCl_i_ (i = 1–4) and CdOHCl complexes were taken from our previous paper [20].

For uranyl(VI), the hydrolytic constants were taken from our previous investigation [21]; moreover, since the UO_2_(Acetate)_2_ salt was used, the speciation and the formation constants of the UO_2_/Ac system in NaCl were considered [22].

For Cu^2+^, the hydrolytic constants were taken from Ref. [23]

### 2.2. Cd^2+^/Dop^−^ System

As already carried out in previous papers on the interaction of biological active molecules with cations, the best speciation model and the corresponding stability constants were selected by using some general rules and guidelines [24,25].

For the cadmium(II) interactions with dopamine, the speciation model, used as input in the BSTAC program [26], includes the protonation constants of the ligand, the hydrolytic species of the metal, the CdCl_i_ (i = 1–4) and the CdOHCl species formed with the ion of the supporting electrolyte (i.e., NaCl), that in some conditions (i.e., high chloride concentration) reach formation percentages higher than 30% [20].

The interactions of dopamine with cadmium(II) were investigated in a quite wide range of experimental conditions, namely *I* = 0.15 mol dm^−3^, from *T* = 288.15 to 318.15 K. Measurements at *T* = 298.15 K were also carried out at different ionic strengths, from *I* = 0.15 to 1.0 mol dm^−3^, and at different metal–ligand molar ratios (more details are reported in the Section 4).

Table 1 reports the stability constants of the Cd*^2+^*/Dop^−^ species at different experimental conditions; the speciation model is formed by only the ML, MLH and ML_2_ (M = Cd^2+^; L = dopamine; charge omitted) mononuclear species. Analyzing the data in Table 1, a difference of about 2 orders of magnitude can be observed between the values at *T* = 298.15 and 310.15 K.

Considering the distribution diagrams reported in the Figure 1, Figure 2 and Figure 3, some considerations can be done:

The importance of the CdCl_i_ species is highlighted in Figure 1, where the distribution and the formation percentages of the species are calculated at *I* = 0.15 mol dm^−3^ and *T* = 298.15 K at metal–ligand molar ratio of 1:2, but similar results were also obtained at the other experimental conditions.

The CdCl^−^ species reaches about 60% of formation at pH values lower than 6, contributing to a reduction of the amount of free Cd^2+^. The other Cd^2+^/chloride species are formed in lower amounts and in dependence on the chloride concentration (i.e., ionic strength) in solution. It is also interesting in the presence of the ternary CdOHCl species that at pH ~ 8 it reaches the maximum of formation.

Up to pH ~ 7, the interaction of dopamine with cadmium does not occur; over this pH, the formation of the MLH species (~10% at pH ~ 8.5) was observed, whilst the ML and ML_2_ ones are formed at higher pHs and in higher amounts, ~80%.

Figure 1 was used as a comparison to evidence by means of Figure 2 and Figure 3, the effect of *I*/mol dm^−3^ and *T*/K on the distribution of the Cd^2+^/Dop^−^ species.

### 2.3. Cu^2+^/Dop^−^ System

The speciation of the Cu^2+^/Dop^−^ system is much more complex with respect to the corresponding system with Cd^2+^. In fact, in this case, the best results were obtained when the formation of the ML_2_, M_2_L, M_2_L_2_, M_2_L_2_(OH)_2_, M_2_LOH and ML_2_OH were considered. In this case, we observed the prevalence of simple binary and ternary hydrolytic binuclear complexes. On the contrary of what obtained for Cd^2+^, the formation of the ML and MLH species does not occur.

Table 2 reports the formation constants of the species obtained at the different experimental conditions (i.e., ionic strengths and temperatures). In the case of the Cu^2+^/Dop^−^ system, we did not observe the formation of precipitate, but measurements were stopped at pH ~ 10, to avoid the eventual oxidation of the ligand.

The interaction between Cu^2+^ and dopamine begins at pH ~ 4, where the formation of the M_2_L species occurs; in particular, the distribution diagram reported in Figure 4 evidence that all the complex species of the Cu^2+^/Dop^−^ system reach significant formation percentages in the pH range between 4.5–10.0. Moreover, the same diagram shows the effect of the ionic strength on the formation percentage of the species. This effect is different for each complex, and, for the CuL_2_, Cu_2_L_2_, Cu_2_L_2_(OH)_2_, the percentage of formation decreases, increasing the ionic strength, while in the case of the Cu_2_L and Cu_2_LOH species, an opposite effect was observed. For the CuL_2_OH species, the same formation percentage (50% at pH = 8.5) was obtained at the two experimental conditions, but we observed a shift versus low pH (pH ~ 7.5) at *I* = 1.00 mol dm^−3^.

Concerning the dependence of the formation percentages on the temperature, information can be obtained from the analysis of Figure 5, where the distribution of the species at *I* = 0.5 mol dm^−3^ and different *T*/K has been drawn. For some of them, a significant variation on the formation percentages was observed, such as for the ML_2_ that at *T* = 288.15 K achieves about the 85% and at *T* = 318.15 K, only the 8%. Similar behaviour has been observed for the M_2_L_2_, whilst in the case of M_2_L_2_(OH)_2_ and M_2_LOH, the formation percentages at *T* = 318.15 K do not exceed the 5%.

### 2.4. UO_2_^2+^/Dop^−^ System

Investigation of the interaction between uranyl(VI) and dopamine resulted different with respect to the other two systems, in part due to the different acid-base behavior of UO_2_^2+^ with respect to Cd^2+^ and Cu^2+^, and to the use of the UO_2_(Acetate)_2_ product for our studies.

For this reason, the speciation model used as input in the BSTAC program [26] contained, the protonation constants of dopamine [19], the hydrolytic species of uranyl(VI) in NaCl(aq) [21] and the formation constants of the UO_2_^2+^/Ac^−^ system [22]. By using the criteria of selection already reported, different mononuclear and binuclear species were tested, but the best results were obtained when the ML_2_ and MLOH species were introduced in the speciation model. With respect to other systems involving the interaction of UO_2_^2+^ towards different classes of organic ligands [22,24,25], in this case the formation of protonated, hydrolytic or polynuclear species was not observed. Moreover, a significant improvement of the statistical parameters was obtained when the MLAc (Ac = acetate) was introduced in the speciation model, together with the other two species, as reported in Table 3 where the stability constants of the complexes at different ionic strengths and temperature conditions were also reported.

The presence of this ternary complex can be explained considering that at the beginning of the potentiometric titrations, the metal is mainly present as acetate salt, forming species of significant stability along the pH interval of investigation where there is simultaneously the interaction with dopamine.

In fact, as reported in the distribution diagrams of the species (Figure 6 and Figure 7) at different ionic strengths and temperatures, we observed that up to pH ~ 5–5.5, the MLAc can be considered as the main species in solution, due also to the high stability of the complex, that has a logβ value of ~16–17.

To better evaluate the stability of this complex, we can calculate the stepwise formation constants considering the following equilibrium of formation: MAc + L = MLAc, where M = UO_2_^2+^, Ac = acetate and L = dopamine (charge omitted). At *I* = 0.15 mol dm^−3^ and *T* = 298.15 K, we have: logK_MAc_ = 2.44 [22] and logβ_MLAc_ = 16.14, for the formation of the logK_MLAc_ = 16.14–2.44 = 13.70. The strength of the complex formed by the interaction of uranyl(VI) with dopamine justifies the absence of possible other species with a higher stoichiometric coefficient.

As already observed for the other two systems, from the distribution diagrams we observed a significant shift of the maximum of formation changing the experimental conditions (i.e., ionic strength and temperature). In the case of the UO_2_^2+^-Ac/Dop^−^ system, the ML_2_ and MLAc species can be considered the main ones, since the MLOH is always formed in lower amounts and over pH ~6.5. The effect of the variables *I*/mol dm^−3^ and *T*/K on the stability of the UO_2_^2+^-Ac/Dop^−^ species can be observed from the Appendix A reported in the Appendix A.

### 2.5. Mixed Dopamine Systems

This group studied the formation of MLL’ and MM’L (M an M’ = generic cations; L and L’ = generic ligands) ternary species for many different classes of organic ligands and metals, and here, for simplicity, only a few papers are cited [27,28,29,30]; the results obtained from those investigations highlighted how an extra-stability contributes to the formation of mixed species, resulting also in some cases in a significant shift of the formation of sparingly soluble species towards higher pH values. Moreover, the same works show that the mixed ternary species tend to form in high formation percentages also at low component concentrations, inhibiting the hydrolysis of metals. For this reason, for a correct speciation study in multicomponent systems, the formation of possible ternary species cannot be neglected. The formation of dopamine ternary mixed complexes was already studied by other authors, but investigations regard only MLL’ mixed ligands systems [31,32,33,34,35,36].

In two previous papers [29,30], the attention was focused on the evaluation whether, in the case of the presence of two metals in solution, the formation of ternary hydrolytic species could be also possible. These investigations concerned the UO_2_^2+^/Cu^2+^ and UO_2_^2+^/Cd^2+^ systems.

The obtained results confirmed the possible formation of ternary M_p_M’_q_(OH)_r_ hydrolytic species. As a further contribution to this kind of study, in this work, an investigation on the UO_2_^2+^/Cu^2+^/Dop^−^ and UO_2_^2+^/Cd^2+^/Dop^−^ systems was performed at *I* = 0.15 mol dm^−3^ and *T* = 298.15 K and different components concentrations, as reported in Section 4.

The speciation models used as input are characterized by many different species, 28 and 32 for the UO_2_^2+^/Cu^2+^/Dop^−^ and UO_2_^2+^/Cd^2+^/Dop^−^ systems, respectively. As an example, in the case of the UO_2_^2+^/Cd^2+^/Dop^−^ systems, the speciation model consists of the following species (in parenthesis number of species): ionic product of water (1); dopamine protonation constants (2); Cd^2+^ hydrolysis (6); CdCl_i_ complexes (5); UO_2_^2+^ hydrolysis (5); acetate protonation constant (1); UO_2_^2+^/Ac^−^ complexes (4); UO_2_^2+^/Cd^2+^ hydrolysis (2); UO_2_^2+^/Dop^−^ species (3); Cd^2+^/Dop^−^ species (3).

The proposed speciation models reported in Table 4 and the overall formation constants of the mixed species were selected based on the criteria of selection for the best speciation model already described. The speciation models have two common species, the MM’L and M_2_M’L_2_OH ones (M = UO_2_^2+^; M’ = Cd^2+^ or Cu^2+^, L = dopamine), but are characterized by different stability, about 4 orders of magnitude.

Generally, when the formation of mixed species is observed, the calculation of the “extra-stability” of formation can be obtained by using the formation constants of the corresponding homo-polynuclear complexes, as discussed by Beck and Nagypàl [37] for the statistical prediction of the stability of mixed complex species.

However, in our case, the stoichiometry of the mixed species does not correspond to the stoichiometry of the homo-polynuclear complexes, and for this reason, the only way to estimate the stability of this species is to calculate the corresponding stepwise formation constants; as an example, this calculation can be done for the mixed metal species reported in Table 5, where the proposed reaction of formation of the mixed species are given.

From the analysis of the distribution diagrams reported in Figure 8 and Figure 9, some important aspects can be highiligthed; (i) the first one is that, as already observed from the other similar investigations, the mixed species form in very high amounts and are in each case the main complexes in the pH ranges investigated; (ii) the strength of the mixed complexes avoid, in the Cd^2+^ containing system, the formation of the hydrolytic species both for UO_2_^2+^ and Cd^2+^ and of the corresponding mixed hydrolytic species; (iii) this behaviour does not occur in the mixed system containing Cu^2+^, where we observed the coexistence of the (UO_2_)_2_(OH)_2_ and (UO_2_)_3_(OH)_5_ species, the reach being about 6–8%; (iv) the formation of the species of the binary systems is avoided by the formation of the mixed ones, except for the UO_2_LAc complex that in each case is formed in a significant amount, starting from pH ~ 3, and reaching about the 10% and 20% of formation in the mixed containing Cd^2+^ and Cu^2+^ systems; (v) as already observed for the binary UO_2_^2+^/Dop^−^ system, also for the mixed ones, the use of the chemical UO_2_(Ac)_2_ salt has a great influence on the formation and distribution of the binary and ternary complexes.

It is important to stress that the distribution of the mixed species and the corresponding formation percentages are related to the component concentration and metal–metal’-ligand (M-M’-L) molar ratios; as an example, in the distribution diagram reported in Figure 8, the absence of the MM’_2_L_2_(OH)_2_ (M = UO_2_^2+^; M’ = Cd^2+^) can be observed, but if the experimental conditions are changed with respect to the one reported in Figure 8, for example at c_UO_2_^2+^_ = 4.0 mmol dm^−3^; c_Cd^2+^_ = 6.0 mmol dm^−3^; c_Dop−_ = 10 mmol dm^−3^, the MM’_2_L_2_(OH)_2_ species reaches about the 54% of formation at pH = 5.4.

### 2.6. Dependence of the Stability Constants on the Ionic Strength and Temperature

The modelling of the formation constants with respect to the ionic strength and the temperature was carried out by using both the extended Debye–Hückel equation and the SIT approach [38,39,40]; further details are given in their dedicated section.

In particular, for Cu^2+^ and UO_2_^2+^ systems, the modelling was carried out by using Equation (3), which allowed us to account for the non-linear variation of the formation constants with respect to *I*/mol dm^−3^ and *T*/K.

By using this approach, it was also possible to calculate the formation constants at infinite dilution, the parameter for the dependence on the ionic strength and the standard enthalpy change values for the formation of the species.

For the Cd^2+^ system, a different modelling approach was applied, in dependence on the different experimental conditions of the measurements, namely: (1) different temperatures and *I* = 0.15 mol dm^−3^; (2) different ionic strengths and *T* = 298.15 K.

The standard formation enthalpy change values were calculated at *I* = 0.15 mol dm^−3^ by using the Van’t Hoff equation:

The formation constants at infinite dilution, the parameters for the dependence on the ionic strength and the standard formation enthalpy change values are reported in Table 6, Table 7 and Table 8.

It is possible to observe that for Cd^2+^ and Cu^2+^, the formation of the species is strongly exothermic, except for the M_2_LOH species of the Cu^2+^/Dop^−^ system. Similar behavior was also observed for the formation of all the species of the UO_2_^2+^-Ac/Dop^−^ system, which resulted endothermic processes. For the application of the SIT approach, the data concentration/ionic strength and formation constants were converted from the molar to molal concentration scale.

The formation constants in the molal concentration scales are reported in Appendix A. In the first instance, the SIT approach expressed by means of Equation (6), in order to calculate the Δε parameters was applied. The formation constants at infinite dilution expressed in the molal concentration scale and the Δε parameters are reported in Table 9, Table 10 and Table 11 for each investigated temperature.

Analysing the results obtained by appliyng the SIT approach expressed by means of Equation (6), a decreasing trend of the Δε values of the Cu^2+^/Dop^−^ species was observed, as better highlighted in Appendix A.

If all the interactions between the ionic component/species with the ions of the supporting electrolyte are known, it is possible to apply the SIT approach expressed by means of Equations (6)–(8) and with the example reported in Equations (9)–(11). For the Cd^2+^/Cl^−^ ion-pair, the specific ion interaction parameter is unknown, and considering negligible in many cases the difference that occurs between the ε(M^2+^; Cl^−^) and the ε(M^2+^; NO_3_^−^) values for the same metal, we decided for Cd^2+^ to use the value of its specific ion coefficient obtained from the interaction with nitrate anion [41]. The specific ion interaction coefficients of the other ionic species present in the solution, namely: ε(H^+^, Cl^−^); ε(Dop^−^, Na^+^); ε(Cd^2+^,NO_3_^−^); ε(Cu^2+^,Cl^−^) and ε(UO_2_^2+^,Cl^−^) must be considered [19,22,41].

For the MLAc species, two different approaches were applied to calculate the Setschenow coefficients of the neutral species; with the first approach, the equilibrium: UO_2_^2+^ + L^−^ + Ac^−^ = UO_2_LAc^0^ was considered and for dioxouranium(VI) the literature-specific ion interaction coefficient ε(UO_2_^2+^, Cl^−^) = 0.25 was used [22,41]. In the second one, the equilibrium and the coefficient considered were: UO_2_Ac^+^ + L^−^ = UO_2_LAc^0^ and ε(UO_2_Ac^+^, Cl^−^) = 0.01 [22]. The Setschenow coefficients are: *k*_mMLAc_ = 1.00 ± 0.07 and 0.67 ± 0.07, respectively. Table 12 reports the ion interaction parameters of the ionic species of the three metal–dopamine systems investigated.

### 2.7. Sequestering Ability of dopamine towards Cations and Effect of pH, Ionic Strength and Temperature

Using Equation (15), the sequestering ability of dopamine towards the metal cations was quantified by means of the determination of pL_0.5_ parameter [20,24,25,44] at different experimental conditions (*I*/mol dm^−3^, pH, *T*/K), as reported in Table 13, Table 14 and Table 15 and in Figure 10, Figure 11 and Figure 12.

As observable in Table 13, Table 14 and Table 15 and in Figure 10, Figure 11 and Figure 12, the pL_0.5_ parameter tends to increase, increasing the ionic strength, temperature, and pH; whilst for the first two variables the increase of pL_0.5_ is quite small, for pH a higher increment can be observed, especially for Cu^2+^ and UO_2_^2+^, where between pH ~ 6.0–9.5 (Cu^2+^) and 5.0–7.4 (UO_2_^2+^), values of ΔpL_0.5_ ~6 order of magnitude can be calculated. In the case of Cd^2+^ the variation observed is of ~2.5 order of magnitude.

The increase of the pL_0.5_ with pH is bound to the deprotonation of the ligand and is an indication of the electrostatic nature of the interactions. However, the results here obtained for the variation of the pL_0.5_ parameter with respect to the variables *I*/mol dm^−3^, *T*/K and pH, are perfectly consistent with those already observed for the interaction of dopamine with Ca^2+^, Mg^2+^, CH_3_Hg^+^ and Sn^2+^ [12].

The different effect of the variables on the pL_0.5_ is better highlighted in the diagrams reported in Appendix A.

Moreover, independent of the variable considered, the sequestering ability trend of dopamine towards the metals ions here investigated, is:pL_0.5_: UO_2_^2+^ > Cu^2+^ > Cd^2+^

In the Literature data section, an accurate comparison with the already published data will be carried out.

### 2.8. Literature Data

Many papers have been published in literature on the chemistry of dopamine and its biological role; however, in many cases, the information reported are not sufficient to understand its behavior in aqueous solutions containing electrolytes that can simulate the biological fluids; in fact, investigations have been generally carried out in pure water or in mixed solvent media [45,46,47,48,49,50,51]. The same problems also subsist for the investigation on the interaction with metal cations.

In a recently published work on the sequestrating ability of dopamine towards other metal cations (Ca^2+^, Mg^2+^, CH_3_Hg^+^ and Sn^2+^), an analysis of the literature data on ligand speciation was proposed [12]. From the analysis of the literature data, it was highlighted that the information is generally valid at a single value of ionic strength and temperature, and very often in aqueous solutions containing supporting electrolytes that do not adequately simulate biological fluids.

Furthermore, the difficulties in comparing the results reported in the different papers are also related to some other factors, such as the neglecting of the effect of metal hydrolysis on the speciation of the metal-dopamine system, or the consideration dopamine as diprotic rather than a triprotic ligand [31,32,33,34,46,52,53,54,55,56,57,58,59].

Moreover, in the case of Cd^2+^ and investigations performed in NaCl ionic medium, the stability of the complexes formed by the the metal ion with Cl^−^ does not allow to neglect the CdCl_i_ (i = 1–4) and CdOHCl species in the speciation model [20], as highlighted in Figure 1 of the present manuscript.

In this work, an upgrade of the literature data table (Table 16) already reported in Ref. [12], containing the literature stability constants of the complexes between Cd^2+^ and Cu^2+^ and dopamine, was proposed. There is no evidence, however, of literature data for the complexes with UO_2_^2+^ in aqueous solution.

Concerning Cd^2+^ and Cu^2+^, only simple mononuclear species were reported, and considering the stoichiometry of the species, it seems that investigations were limited only to the acid pH range, especially for Cu^2+^, that tends to form ternary hydrolytic species [31,60,64,65].

In the case of Cd^2+^, authors reported only a single Cd^2+^/Dop^−^ species, namely the MLH one.

Considering that the literature reports different speciation models for the various metal–ligand systems, in turn determined in different experimental conditions, it is particularly difficult to make a comparison based only on the formation constants.

The most suitable method in these cases is to use the pL_0.5_, described in the Section 4.4. and in many previous works, since this parameter is independent of the speciation model when the metal is present in traces, and it is only dependent on the experimental conditions.

By using the available literature and experimental data, it was possible to draw a sequestering diagram that can allow us to evidence the effective ability of dopamine to interact with the metal cations (at a given experimental conditions), considering in the speciation model of each system, the protonation constants of dopamine, the hydrolytic constants of the metal, the possible stability constants of the species formed by the interaction of the metal with the anion of the supporting electrolyte (namely Cl^−^ for this work) and the stability constants of the metal/dopamine species. This diagram was drawn at *T*~298.15 K and *I*~0.15–0.2 mol dm^−3^, and the results are reported in Figure 13. Considering the results in Figure 13, the following sequestering trend was obtained, where the pL_0.5_ values for Mg^2+^ and Ca^2+^ were reported, but not inserted in the diagram:

pL_0.5_: Mg^2+^ (0.08) < Pb^2+^, 0.23 < Ca^2+^ (1.15) < Mn^2+^, 1.61 < Ni^2+^, 1.77 < Zn^2+^, 2.61 < CH_3_Hg^+^, 2.63 < Sn^2+^, 3.67 < Cd^2+^ (pH = 9.5) 3.80 < DET (CH_3_CH_2_)_2_Sn^2+^, 4.53 < Cu^2+^, 4.64 < UO_2_^2+^, 7.58.

From the analysis of Figure 13, some aspects can be evidenced; the first one is that dopamine has, in the same experimental conditions, a significantly different in the sequestering ability toward the different metal ions.

This behaviour is due to the different acid-base properties of the metal ions that tend to hydrolyze in different pH ranges and form mono- and/or polynuclear species of different stabilities, reducing proportionally the amount of free metal able to interact in those conditions with the ligand. Another factor that influences the sequestering ability and the amount of free metal is the possibility to form complexes with the anion of the supporting electrolyte stable, often stabilized by the amount of salt in solution.

For this reason, the pL_0.5_ values calculated for each metal in Figure 13 are valid only at those conditions; changing them, different pL_0.5_ and also different trends, can be obtained for the sequestering ability.

The trend of sequestration reported is mainly consistent with the stability of the complexes. The pL_0.5_ value calculated for Pb^2+^ seems to be anomalous. This value can be justified taking into account the experimental conditions where it was calculated, and in particular the pH value (7.4). Since the literature reports for the interaction of Pb^2+^ with dopamine only the MLH species that forms mainly at pH < 6, at pH = 7.4, the amount of complexed ligands by the metal is negligeble, at about 2%.

## 3. Discussion

The speciation studies carried out on dopamine interactions towards binary e ternary UO_2_^2+^, Cu^2+^ and Cd^2+^/dopamine systems lead to the following conclusions:quite different speciation models were obtained for the three different binary systems.The speciation model for the Cd^2+^ is featured by mononuclear complexes (i.e., MLH, ML and ML_2_ species) that cover the pH range were investigated; in this case, the stability and distribution of the species is also dependent on the CdCl_i_ species formed with the anion of the supporting electrolyte.For Cu^2+^, the system is particularly complex owing to the presence of different binuclear (M_2_L and M_2_L_2_), binary hydrolytic (M_2_LOH, M_2_L_2_OH_2_) and mononuclear (ML_2_ and ML_2_OH) species.For UO_2_^2+^, the investigations were carried out by using the UO_2_(Ac)_2_ salt; this aspect influenced the speciation, because as already observed in other papers, uranyl(VI) tends to form polynuclear complexes with organic ligands, like dopamine; here we observed the formation of the ML, MLOH and the mixed ternary MLAc species, that are formed in high amounts at a wide pH interval.Measurements carried out at different ionic strengths and temperatures allowed us to observe that the stability of the complexes is influenced by changes of these two variables, and that the entropy is generally the driving force of the metal/dopamine species formation.The dependence of the stability constants on the ionic strength and temperature was modelled by means of the extended Debye–Hückel equation that allowed the calculation of the stability constants at infinite dilutions, the parameters for their dependence on *I*/mol dm^−3^ and *T*/K, with the determination of enthalpy change values.The stability constants dependence on ionic strength (expressed in the molal concentration scale) was also modelled by means of the Specific ion Interaction parameters (SIT) that allowed the calculation of the specific ion interaction parameters and of the Setschenow coefficient for the neutral species formed by the interaction of the metals with dopamine.The results obtained from the investigations carried out on the mixed systems highlighted that they cannot be neglected in a correct speciation study of multicomponent fluids, where the contemporary presence of metals and ligands can lead to the formation of mixed complexes of high stability, which tends to avoid or reduce the percentage of formation of the binary species, influencing the speciation of the whole system and the distribution of the species at a given experimental condition.By using the pL_0.5_ parameter, it was possible to quantify, at different pHs, ionic strengths and temperatures, the effective sequestering ability of dopamine towards the metals. This approach is very important, since a simple comparison of the stability constant values can lead to incorrect considerations, especially when comparing systems with different speciation and in different experimental conditions of temperatures, ionic strengths, pHs or in different ionic media. The results here obtained highlight that the sequestering ability tends to vary as the temperature and ionic strength change. The sequestering ability of dopamine towards the metals follows the trend:pL_0.5_: UO_2_^2+^ > Cu^2+^ > Cd^2+^An analysis of the literature data, already undertaken in Ref. [12] and here updated, allowed to observe the different speciation models and stability of the metal/dopamine complexes, and that a possible comparison in terms of ability of dopamine to interact with the different metal ions can be obtained by using the pL_0.5_ parameter. From the results above reported, it is possible to observe that the higher sequestering ability is towards uranyl(VI), followed by Cu^2+^.

## 4. Materials and Methods

### 4.1. Chemicals

For the investigation carried out in this work dopamine hydrochloric salt was used; the solutions were prepared by weighing the chemical without further purification. The purity of the chemical was checked potentiometrically by alkalimetric titrations and resulted to be >99.5%. Sodium chloride aqueous solutions at different ionic strengths were prepared from the dilution of a concentrated solution, obtained by weighing pure salt previously dried in an oven at *T* = 383.15 K for 2 h. Sodium hydroxide and hydrochloric acid solutions were prepared from concentrated ampoules and were standardized against potassium hydrogen phthalate and sodium carbonate, respectively.

Standard stock solutions of the metals (Cu^2+^, Cd^2+^, UO_2_^2+^) were prepared from the corresponding chloride salts and were used without further purification.

For Cu^2+^ and Cd^2+^, the concentration of the metal ions in the aqueous solutions were determined by means of complexometric titrations with EDTA (Ethylendiaminetetraacetic acid sodium salt) [66]. For UO_2_^2^^+^, diacetate salt was used, and the purity was determined through the gravimetric determination of uranium after ignition to the oxide U_3_O_8_ [22].

All products were purchased from Sigma-Aldrich, Milan, Italy. All solutions were prepared with analytical grade water (ρ = 18 MΩ cm^−1^) using grade A glassware and were preserved from atmospheric CO_2_ by means of soda lime traps.

Further details on the chemicals used for the investigation here carried out are reported in Table 17.

### 4.2. Apparatus and Procedure

The investigations of the interactions of dopamine towards Cu^2+^, Cd^2+^, UO_2_^2+^ were carried out by means of potentiometric titrations by using an apparatus consisting of an 809 model Metrohm Titrando system (Metrohm, Varese, Italy), connected to a half-cell Ross Type glass electrode (model 8101 from Thermo Fisher Scientific, Waltham, MA, USA), coupled with a standard Ag/AgCl reference electrode. The apparatus also consists of a personal computer that, by means of the installed Metrohm TiAMO 2.2 computer program (Metrohm, Varese, Italy), allows the performance of automatic titrations, by the addition of the desired amounts of titrant when the equilibrium state is reached, to control the parameters of the titrations and to record the e.m.f. of the solution. The estimated accuracy was ±0.15 mV and ±0.003 mL for e.m.f and titrant volume readings, respectively.

The measurements were carried out under continuous stirring and pure N_2_ flow in a thermostated cell, connected to a thermocryostat (Model D1-G-Haake, Gebrüder HAAKE GmbH, Karlsruhe, Germany), by means of water circulation in the outer chamber of the titration cell. Investigations were carried out from *T* = 288.15 to 310.15 K.

The titrated solutions consisted of different amounts of the cation, dopamine⋅HCl, an excess of hydrochloric acid and NaCl, for obtaining the desired ionic strength values. To investigate the possible formation of both mono- and/or polynuclear species, solutions were prepared in a wide range of metal ion to ligand molar ratios and were titrated with standard carbonate-free NaOH, up to alkaline pH values or to the formation of sparingly soluble species. Table 18 and Table 19 report the experimental conditions employed in the investigations of the binary (components concentration, ligand–metal molar ratios) and ternary MM’L (M = UO_2_^2+^; M’ = Cu^2+^ or Cd^2+^; L = dopamine) systems, respectively.

For each titration, the total number of potentiometric experimental points collected varied between 60–100, in dependence on the possible formation of a sparingly soluble species. At least two measurements were carried out for each experimental condition. Independent titrations of strong acid (HCl) solutions with NaOH solutions were carried out at the same experimental conditions of the systems investigated, with the aim of determining the electrode potential (E^0^) and the acidic junction potential (Ej = ja [H^+^]).

In this way, the pH scale used was the free concentration scale, pH ≡ −log [H^+^], where [H^+^] is the free proton concentration (not activity). The reliability of the calibration in the alkaline range was checked by calculating the ionic product of water (pK_w_).

Figure 14 reports the titration curves of dopamine and of the metal/dopamine systems at *I* = 0.15 mol dm^−3^ and *T* = 298.15 K, where it is possible to observe quite different profiles of the curves, depending on the different acid-base properties of the metal ions.

The different aspect of the UO_2_^2+^/Dop^−^ titration curve, quite different with respect to the others, is due to the use of UO_2_(Acetate)_2_ salt instead of the UO_2_(NO_3_)_2_ one, generally used in similar investigations. For CH_3_Hg^+^, Ca^2+^, Mg^2+^ and Sn^2+^, the results of the investigation on the interaction towards dopamine were recently published [12], whilst the studies with DET [(CH_3_CH_2_)_2_Sn^2+^] are in progress.

### 4.3. Calculations and Models for the Dependence of the Stability Constants on Ionic Strength and Temperature

#### 4.3.1. Computer Programs

All the parameters of the potentiometric titrations (standard electrode potential (E^0^), liquid junction potential coefficient (j_a_), ionic product of the water (K_w_), analytical concentration of reagents and formation constants) were determined by using the non-linear least-square minimization method and the BSTAC program.

By using the BSTAC program, the error square sum in the electromotive force (E = e.m.f.) was minimized:U = ∑W·(E_exp_ − E_calcd_)^2^
(1)

Further details on the BSTAC computer program are reported in Ref. [26].

The LIANA (LInear And Nonlinear Analysis) computer program [26], which minimizes the error square sum in y of an equation y = f(x_i_) (a generic function given by the user), was employed for the determination of the equilibrium constants at infinite dilutions and the corresponding parameters for the dependence on the ionic strength and of the Specific ion Interaction Theory parameter of the ion-pairs.

The HySS program [67] was used for the calculation of the formation percentages of the species present in solution at the equilibrium and to draw the distribution diagrams in different conditions. The HySS program allows to consider the formation of the sparingly soluble species, by using the solubility product of the species in the speciation model.

Within the manuscript, if not differently specified, hydrolysis (q = 0, r < 0) constants of cations, protonation (p = 0) constants of the ligands (L^z−^) and complex formation constants are given according to the overall equilibrium:logβ_pqr_: p M^n+^ + q L^z−^ + r H^+^ = M_p_L_q_H_r_^(np−zq+r)^(2)

The errors associated with formation constants, standard enthalpy and entropy change values and parameters for the dependence on ionic strength, are expressed as ± standard deviation (Std. Dev.).

#### 4.3.2. Dependence of the Stability Constants on the Ionic Strength and Temperature

The constant ionic medium method, which consists of adding in the solution a supporting electrolyte at concentration higher with respect the components, for different order of magnitude was used for the investigations. In this order, it is possible to assume that the activity coefficient of the components have a unitary value so that the activity ≈ concentration [68].

The dependence of the formation constants on ionic strength was studied both by means of the Extended Debye-Hückel (EDH) equation and Specific ion Interaction Theory (SIT) approach [38,39,40].

#### 4.3.3. Extended Debye-Hückel (EDH)

The dependence on the ionic strength and temperature of the complex formation constants was modelled by means of an extended Debye-Hückel (EDH) equation:logβ_pqr_ = logβ^T^_pqr_ − z* × A × √*I*/(1 + 1.5√*I*) + C∙*I* + L∙(1/298.15 − 1/*T*) × 52.23 (3)
where 52.23 is 1/(R∙ln10) and C is the adjustable parameter for the dependence of formation constants on ionic strength in the molar concentration scale, composed as follows: C = c_0_ p* + c_1_ z*; p* = ∑p_reactants_ − ∑p_products_ and z* = ∑z^2^_reactant_ − ∑z^2^_product_, where z and p are the charge and the stoichiometric coefficients, respectively. Logβ^T^_pqr_ is the formation constant at infinite dilution and A is the Debye-Hückel term, whose empirical parameters for the dependence on the temperature are reported in Equation (4), where *T* = 298.15 K is the reference temperature and *T*’ is the desired one, expressed in kelvin (K):A = (0.51 + (0.856·(*T’* − 298.15) + 0.00385·(*T* − 298.15)^2^)/1000)(4)

The term L, valid when measurements at different temperatures are carried out, allows the calculation, in the investigated ∆*T* range, of the standard enthalpy change value of formation of a given species at infinite dilution ∆H_n_^0^.
L = (∆H_n_^0^ − z* × (1.5 + 0.024 ∙ (*T’* − 298.15)√*I*)/(1 + 1.5√*I*))(5)

The Equation (5) report also the term for the dependence of the ∆H_n_^0^ on the ionic strength (*I*/mol dm^−3^).

For the Cd^2+^/Dop^−^ system, since the dependence of the stability constants on the ionic strength was only investigated at *T* = 298.15 K, the Debye-Hückel (EDH) equation used is like the one reported in Equation (5), except that for the L term for the dependence on *T*/K. The standard formation enthalpy change values of the complexes were calculated by using the classical Van’t Hoff equation.

#### 4.3.4. Specific ion Interaction Theory (SIT) Approach

If the ionic strength and the stability constants are converted from the molar to the molal concentration scale, Equation (3) becomes (neglecting the last term containing ∆*H*_n_^0^) the classical SIT (Specific ion Interaction Theory) equation, in which C is replaced by ∆ε, as follows:logβ_pqr_ = logβ^T^_pqr_ − z* × (A√(*I*_m_))/(1 + 1.5√*I*_m_) + ∆ε × *I*_m_ + j × loga_w_(6)
where *I*_m_ = ionic strength in the molal concentration scale, log a_w_ is the activity coefficient of water (log a_w_ = 0.015), j = number of water molecules involved in the equilibrium.
∆ε = ∑ε_reactants_ − ∑ε_products_
(7)
ε is the SIT coefficient for the interaction of the ionic species involved in the considered equilibrium with the ion (of opposite sign) of the ionic medium. For neutral species, the SIT coefficients are expressed by means of the Setschenow equation [43].
log γ = k_c,m_ × *I*(8)
where k_c_ and k_m_ are the Setschenow coefficients of the neutral species in a given medium, in the molar and molal concentration scales, respectively.

The ∆ε parameter of Equation (6) can be explicated to obtain the ion-pairs SIT coefficients for all the species involved in the equilibrium of formation of the complexes; as an example, for the Cd^2+^ ion, the specific ion interaction coefficients can be calculated for the ML species as follows:∆ε = ε(M^2+^,Cl^−^) + ε(L^−^,Na^+^) − ε(ML^+^,Cl^−^)(9)

For the MLH species:∆ε = ε(M^2+^,Cl^−^) + ε(L^−^,Na^+^) + ε(H^+^,Cl^−^) − ε(MLH^2+^,Cl^−^)(10)
for ML_2_:∆ε = ε(M^2+^,Cl^−^) + 2 ε(L^−^,Na^+^) − k_m_(ML_2_^0^)(11)
where k_m_ is the Setschenow coefficient of the neutral species.

If a ternary hydrolytic species is formed, the activity coefficient of water must be considered (log a_w_ = 0.015) in the calculation of the specific ion interaction parameter; as an example, in the case of the Cu^2+^/Dop^−^ system, where the formation of the M_2_LOH species was obtained, and whose overall equilibrium of formation can be written as follows:2 Cu^2+^ + L^−^ + H_2_O = Cu_2_LOH^2+^ + H^+^

In this case, the specific ion interaction coefficients can be calculated in the following way:∆ε = 2 × ε(M^2+^,Cl^−^) + ε(L^−^,Na^+^)−ε(M_2_LOH^2+^)–ε(H^+^,Cl^−^)−0.015 *I*/mol kg^−1^(12)

### 4.4. Sequestering Ability of Dopamine toward Metal Ions

By using the pL_0.5_ approach already reported in many previous investigations [20,24,25,44], the sequestering ability of a ligand towards a metal ion at different experimental conditions (ionic strength, ionic medium, pH and temperature) can be quantified. The pL_0.5_ parameter can be calculated by applying a Boltzmann type equation to the couple of data: sum of the mole fraction of all the formed metal–ligand species vs. the minus logarithm of the analytical concentration of the ligand (pL).

The proposed equation is:*x* = 1/(1 + 10^(pL−pL_0.5_)^)(13)
where *x* is the total fraction of complexed metal plotted versus pL (pL = −log [L]_tot_ and [L]_tot_ is the analytical concentration of ligand).

This function is like a sigmoidal curve with an asymptote of 1 for pL → −∞ and 0 for pL → +∞. pL_0.5_ is a quantitative parameter and represents the total concentration of ligand necessary for the sequestration of 50% of the metal cation. In the calculation of pL_0.5_, all parallel reactions (metal hydrolysis, ligand protonation, reactions with other components, etc.) are considered in the speciation model but are successively excluded in the calculation of pL_0.5_.

## Data Availability

All the experimental data are reported in the main text or in supporting files. Any other information about data handling may be obtained upon contacting Francesco Crea (fcrea@unime.it).

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
