# Peer review of "The Effect of Metal Cations on the Aqueous Behavior of Dopamine. Thermodynamic Investigation of the Binary and Ternary Interactions with Cd2+, Cu2+ and UO22+ in NaCl at Different Ionic Strengths and Temperatures"

_molecules, 2021, doi:10.3390/molecules26247679_

Round 1

Reviewer 1 Report

The authors report thermodynamic data for the complex formation reaction of dopamine with 3 metal cations. Huge amount of experimental work is included in the manuscript but the new scientific content and significance of the work is rather low. Major revision of the ms. and spectroscopic evidence are necessary before publication.

Major points for revision:

  1. The manuscript is extremely long as compared to its new scientific content. At least 50% reduction is required.
  2. In the last few decades (from the 1970's) many publications are available in literature for similar systems. In some cases, especially for copper(II), there are significant differences between the old and recent speciation models. Simple potentiometry is not satisfactory to justify the reality of these models. Depending on the metal ion different spectroscopic measurements must be performed to prove the existence of these species.

Author Response

Reviewer 1

The authors report thermodynamic data for the complex formation reaction of dopamine with 3 metal cations. Huge amount of experimental work is included in the manuscript but the new scientific content and significance of the work is rather low. Major revision of the ms. and spectroscopic evidence are necessary before publication.

Concerning this point commented by the referee “…the new scientific content and significance of the work is rather low”, an analysis of the literature data has been done, and the most important results are reported in Table 16. The clear evidence is that:

  1. literature data are completely absent regarding the interaction of dopamine with UO22+; some results, but a single ionic strength and temperature, are reported for Cd2+ and Cu2+, and in many cases the data are not in agreement with each other.
  2. Investigation in literature were generally carried out at an acidic pH range, without considering the possible formation of ternary hydrolytic species, abundantly reported in the literature for similar systems.
  3. In our knowledge, this is the first time that for these systems, a modelling of the formation constants with respect to the ionic strength and temperature has been carried out, also allowing the determination of the enthalpy and entropy of formation of the complexes.

Major points for revision:

  1. The manuscript is extremely long as compared to its new scientific content. At least 50% reduction is required.

We understand the referee's comment regarding the length of the manuscript, but we also want to highlight that the work reports:

  1. results on the investigation on 3 metals / dopamine systems;
  2. each system was investigated at 4 ionic strength values ​​and at 3-4 temperatures (with the exception of Cd2+);
  3. for each system, the dependence on ionic strength was studied by using two different approaches;
  4. the dependence of the formation constants on the temperature allowed the calculation of the enthalpy and entropy change values of formation of the complex species, data absent till now from the literature;
  5. the sequestering ability at different ionic strength values, pH and temperatures was investigated by means of the pL0.5 parameter;
  6. two metal1-metal2 / dopamine ternary systems were also studied;
  7. a comparison with literature data was also done.

The data reported here are certainly sufficient for the publication of 3 independent manuscripts (one for each system), but for intellectual correctness we preferred to report everything in the same manuscript due to the lack of such data in the literature, especially as regards the dependence on strength and temperature.

  1. In the last few decades (from the 1970's) many publications are available in literature for similar systems. In some cases, especially for copper(II), there are significant differences between the old and recent speciation models. Simple potentiometry is not satisfactory to justify the reality of these models. Depending on the metal ion different spectroscopic measurements must be performed to prove the existence of these species.

The referee considerations are partially correct. It is important to highlight that according to the IUPAC, still today, the potentiometric measurements are the most suitable for speciation studies, as no external force, that can alter the equilibria between the components and the species, is imposed on the system, allowing to investigate wide interval of components concentrations and metal:ligand molar ratios, higher with respect to the other techniques.

Moreover, if the spectrophotometric measurements must be performed, lower concentrations with respect to the potentiometric titrations must be used, especially when the given ligand has, as in the case of dopamine, high molar extinction coefficient values.

If the metal-ligand systems tend to form polynuclear species and measurements are performed at low concentration values, there is the risk of not identifying them. In these cases, the spectrophotometric measurements allow the determination of the mononuclear species that are formed in a higher percentage or only some of the polynuclear ones (those formed in higher amount). Our research group used the spectrophotometric measurements for the investigation of the interaction between many different metal-ligand systems, some of them are reported in reference 11, 14-18 of the manuscript, but the spectrophotometric data generally were useful to confirm the potentiometric results, obtaining in many cases formation constants with errors (expressed as Std. Dev.), higher with respect those determined from potentiometry.

Therefore, we disagree with the referee's comment on the necessity of spectrophotometric investigations to validate our results. We want also to highlight that also the large part of the literature results on the metal-dopamine speciation are obtained from potentiometry.

As for the referee's comment on the different speciation models, it probably derives from a careless evaluation, since the various speciation models reported in literature depend on many different variables, such as; different concentrations of the components and relative molar ratios; different pH ranges investigated, etc. In the comment, the referee gives the example of copper; the literature data reported in Table 16 denote that also these values, published from various authors are different with each other both as regards the speciation models and the value of the formation constants. In particular, reading the literature papers, it is noted that the pH range investigated is limited, in many cases, to the acidic range of pH, while our investigations have been carried out up to pH ~ 10, allowing also the determination of the ternary hydrolytic species.

Moreover, we want to highlight that for copper, cadmium and uranyl, the speciation models reported in the manuscript, were obtained by independently processing the potentiometric data at the single ionic strength and temperature investigated (as an example, for Cu2+ and UO22+ at 4 ionic strength values at the same temperature, and 3-4 temperatures). Therefore, we are confident of the correctness of our results.

Reviewer 2 Report

This paper deals with speciation analysis of reaction systems that contain aqueous solutions of the metal ions: Cd(II), Cu(II) and UO22+  and dopamine at different acidity of the media. The factors which were chosen for optimization are ionic strength and temperature.  Since the coordination chemistry of dopamine has long been studied (for example: T. Kiss and A. Gergely, Journal of Inorganic Biochemistry 25, 247-259 (1985)) and the behavior of selected metal ions in aqueous solutions at different acidity is well known, stoichiometry of the metal-dopamine-containing species that can be obtained is quite clear.

The subject of this article should be interesting, given the biological behavior of the compound, but unfortunately the data presented are not new and the analysis is in many cases incorrect. And also there are no conclusive evidences for the composition of chemical species in solutions under investigation and the proposed methods for its determination are not sufficient.

This is the reason why I propose this paper to be rejected.

Some considerations:

  • Most of the results presented in the section “Results” are slightly modified from data that are already published. It is completely unacceptable even in “Supplementary material” to present copied tables with already published data in other articles, especially since they do not provide a further explanation of the claims of the presented study. In general, it is very difficult to separate the new results from the already published ones.
  • It is completely unclear the choice of the interval of values of ionic strength and most important how it is determined - only from the concentration of NaCl or the effect of the starting compounds of metal ions are also included? In this respect it is necessary to give the initial concentration of the latter compounds in part “Experimental” also  (nor only under Figures) because it is well known that the value of ionic strength is determined from all presented ions in solutions and the concentration of some of the electrolytes can be neglected if it is much lower.
  • The application of of the limited Debye–Hückel equation or extended Debye–Hückel equation is determined by the value of ionic strength and also for higher values such as bigger then 0,8 it should be used other equations such as a Devies equation, for example. These equations are used in order to determine the activity coefficients of the ions in solutions,  which mean to fine the relationships between the concentrations of the starting compounds and the activities of their ions in solution. In this respect it is completely clear that at different activities, the composition and the formation constant of the species will be different.
  • It is unacceptable to present the metals, for example copper as Cu in solution, even Cu2+ in water solution below pH~7 exists as [Cu(H2O)6].
  • The presenting of the distribution diagrams as function of pH is unclear. It is accepted these diagrams to present as function of the ligand concentration, but the ligand can be dopamine, OH-, etc. More over in this paper is unclear how the pH value is determined precisely. If equilibrium systems are studied, the use of HCl and NaOH alone is not sufficient and the use of buffer systems is mandatory. Also the parameter pL0,5 is not clearly define.
  • Also there are many notes about the presentation of the equilibria. The experimental conditions used for this study is far away from the biological conditions and only 0,15 mol/L NaCl and 37 C degree are not enough to modeled them and etc.

Author Response

Reviewer 2

This paper deals with speciation analysis of reaction systems that contain aqueous solutions of the metal ions: Cd(II), Cu(II) and UO22+  and dopamine at different acidity of the media. The factors which were chosen for optimization are ionic strength and temperature.  Since the coordination chemistry of dopamine has long been studied (for example: T. Kiss and A. Gergely, Journal of Inorganic Biochemistry 25, 247-259 (1985)) and the behavior of selected metal ions in aqueous solutions at different acidity is well known, stoichiometry of the metal-dopamine-containing species that can be obtained is quite clear.

The subject of this article should be interesting, given the biological behavior of the compound, but unfortunately the data presented are not new and the analysis is in many cases incorrect. And also there are no conclusive evidences for the composition of chemical species in solutions under investigation and the proposed methods for its determination are not sufficient.

This is the reason why I propose this paper to be rejected.

Some considerations:

  • Most of the results presented in the section “Results” are slightly modified from data that are already published. It is completely unacceptable even in “Supplementary material” to present copied tables with already published data in other articles, especially since they do not provide a further explanation of the claims of the presented study. In general, it is very difficult to separate the new results from the already published ones.

The referee's statement seems somewhat strange to us considering different evaluations.

  1. First of all, we want to specify that the supplementary material section gives the possibility to insert a whole series of data and figures in support of the experimental ones reported in the manuscript, including literature data that serve to support them.

In the case of our work, it seems quite clear that the Tables (S1-S6) of the supplementary material section attached to the original manuscript, report: dopamine protonation constants and hydrolytic constants of metals used as input in the speciation models and necessary to process the experimental potentiometric data. In the case of uranyl, also the formation constants with the acetate ion have been considered, and for the mixed systems, also the mixed hydrolyses UO22+ / Cu2 + and UO22+ / Cd2+.

Moreover, if the referee had read more carefully this part, he would have seen that the table caption clearly states ... Literature ............ "

  1. As regards the other Tables (S7-S9), these are not a copy of those reported in the manuscript, but they report the stability constants of the metal / ligand species in the molal concentration scale.

It is well known that at low ionic strength, the conversion of the ionic strength and stability constants from the molar to the molal concentration scale gives in many case results like those here obtained, namely Dlogbpqr values lower than the significant number (here second decimal) of the stability constants. As an example the conversion of the ionic strength and stability constants at I = 0.15 mol dm-3 in NaCl aqueous solution and T = 298.15 K is: I = 0.150 mol dm-3 à I = 0.1508 mol kg-1; logKpqr (mol kg-1) = logKpqr(mol dm-3) – 0.0024; at I = 1.00 mol dm-3 à I = 1.022; logKpqr (mol kg-1) = logKpqr(mol dm-3) – 0.00937

Moreover, we do not understand the affirmation of the referee: “Most of the results presented in the section “Results” are slightly modified from data that are already published”.

We want to point out, that in our knowledge and for the systems here investigated, that there are not data reported in the literature for:

  1. studies performed in these pH intervals, from about 2 to 10;
  2. studies for the dependence of the formation constants on the ionic strength;
  3. studies for the dependence of the formation constants on the temperature;
  4. determination of the enthalpy and entropy change values of formation of the metal:ligand species.

So, we want to stress that the referee comment is unacceptable if it does not specify better what data “have already been published”. In fact, our data are completely different from the literature data reported in Table 16. For example, for the uranyl-dopamine system, there are no data already present in the literature.

  • It is completely unclear the choice of the interval of values of ionic strength and most important how it is determined - only from the concentration of NaCl or the effect of the starting compounds of metal ions are also included? In this respect it is necessary to give the initial concentration of the latter compounds in part “Experimental” also  (nor only under Figures) because it is well known that the value of ionic strength is determined from all presented ions in solutions and the concentration of some of the electrolytes can be neglected if it is much lower.

Regarding this comment, the choice of the ionic strength interval was made considering the necessity to make a thorough study of speciation, to investigate not only the ionic strength value similar to the one of biological fluids (i.e. ~ 0.16 mol dm-3), but also higher values, in this case up to I = 1 mol dm-3. This allowed to model the dependence of the formation constants on the ionic strength, by means of the extended Debye-Huckel equation, and by means of the SIT theory, the calculation of the specific ionic interaction coefficients of the metal-ligand ionic species and of the Setschenow coefficients for the neutral ones.

Concerning the question of the referee on the calculation of the ionic strength, we can replay as follow:

the experimental ionic strength values, reported in Tables 1-4, were calculated considering:

  1. concentration of the supporting electrolyte (i.e. NaCl) in mol dm-3;
  2. initial concentration of the single components (metal ion, dopamine, H+, Cl-, etc.) present in the solution.

Moreover, the BSTAC software used for the determination of the formation constants, allows also to consider the variation of the ionic strength during the titration, taking into account the dilution effect when the titrant was added, the deprotonation of the ligand due the variation of the pH, the contribute to the ionic strength of the metal ion and its hydrolytic ionic species and the formation of the different metal-ligand species. This calculation was carried out for each point of the potentiometric titration, and at the end, the program gave us an average value of I/mol dm-3.

Concerning the referee comment: “In this respect it is necessary to give the initial concentration of the latter compounds in part “Experimental” also  (nor only under Figures)”, perhaps he has read this part of our work too little carefully, since Table 18 (in the case of the mixed systems in Table 19) reports what he asked, namely, the initial concentration of the components (metal and dopamine). More details are also reported in the Section 4.2. Apparatus and procedure. So, this information is not only reported, as he stated, but are also present in the footnote of the figures.

  • The application of of the limited Debye–Hückel equation or extended Debye–Hückel equation is determined by the value of ionic strength and also for higher values such as bigger then 0,8 it should be used other equations such as a Devies equation, for example. These equations are used in order to determine the activity coefficients of the ions in solutions,  which mean to fine the relationships between the concentrations of the starting compounds and the activities of their ions in solution. In this respect it is completely clear that at different activities, the composition and the formation constant of the species will be different.

This referee comment is not very clear, nor what the question really is. Therefore based on our evaluation, we report the following comment.

The considerations made by the referee are partially correct.

We agree with him regarding the limits of the original Debye-Huckel equation in terms of application to high ionic strength values. However, the equation we used to model the dependence of the formation constants on the ionic strength, is an extended equation, widely used in literature, which contains an empirical term, namely C, that accounts for the variation of the formation constants with the ionic strength. This equation is valid up to I = 1.0 mol dm-3. For higher values of I / mol dm-3, it is possible to add further parameters (D I^ 3/2) and (E I^2) which allows the application of the equation up to I ~ 6 mol dm-3.

Concerning the comments on the activity coefficient, we want to remember the reviewer that owing to the difficulties to calculate the activity coefficient and their corresponding variation, the speciation studies are generally carried out by using the constant ionic medium method, that consist of adding in the solution a supporting electrolyte at a concentration higher with respect the components, for different order of magnitude. In this order, it is possible to assume that the activity coefficient of the components have unitary value so that the activity » concentration.

Referee can find more details in: “Chemistry of Marine Water and Sediments”. Editors:  Antonio Gianguzza; Ezio Pelizzetti, Silvio Sammartano (2002) - Springer.

  • It is unacceptable to present the metals, for example copper as Cu in solution, even Cu2+ in water solution below pH~7 exists as [Cu(H2O)6].

It is obvious that when we talk about a metal in solution, we refer to its cation and in particular to its aqueous species; in the case of copper to Cu2+.

Moreover, it seems evident from the information reported in Table 17, that we used the salts of the cations to prepare the solutions, certainly not the metal in the elementary state.

Even the comment of the referee is incorrect, because in the example that he makes for Cu2+ and on the aqueous ion [Cu(H2O)6], we remember that this metal in aqueous solution tends to hydrolyze at pH values much lower than pH ~ 7 that he cites .....

On the other hand, if the referee does a literature study, he will see that the authors of this work, and the research group of prof. Silvio Sammartano, have devoted much of their research to the study of the acid-base properties of metals in solution, and therefore to their cationic (Mn+) species, determining the hydrolysis constants for a large number of metal cations, in different ionic media, at different ionic strength values and temperatures. Moreover, studies performed by means of isoperibolic calorimetry measurements, allowed the calculation of the enthalpies and entropies change values associated with the hydrolysis of the metal cations.

  • The presenting of the distribution diagrams as function of pH is unclear. It is accepted these diagrams to present as function of the ligand concentration, but the ligand can be dopamine, OH-, etc. More over in this paper is unclear how the pH value is determined precisely. If equilibrium systems are studied, the use of HCl and NaOH alone is not sufficient and the use of buffer systems is mandatory. Also the parameter pL0,5 is not clearly define.

For researchers confident with speciation studies, it is well known that species distribution diagrams are used to highlight the pH ranges where the species of a generic metal/organic or inorganic ligand system are formed at fixed metal and ligand (dopamine in this case) concentrations at a given ionic strength and temperature.

The diagram type cited by the referee can only give the different molar fractions of the metal complexed at the different ligand (dopamine) concentration, which are not distribution diagrams.

Regarding the meaning of ligand, at page 4 of the manuscript we reported “…includes the protonation constants of the ligand”, namely dopamine. Moreover, from the footnote of each Figure, it is very clear that when we talk about ligand, we refer to dopamine. Therefore, this statement by the referee leaves us somewhat perplexed.

Also, regarding how the pH was determined, we recommend that the referee reread section 4.2. Apparatus and Procedure. We also want to remind the referee that the distribution diagrams are not experimental, but a simulation of the distribution of the species at a given experimental condition (ionic strength, temperature, component concentrations). These diagrams are drawn considering all the species present in the speciation model and the corresponding stability constants.

For the last referee statement, regarding the need to use a buffer in speciation studies, we want to remind that the use of buffers to maintain fixed the pH within a desired range is a practice in chemical, biochemical and biological studies. However, this approach is very limiting, as it allows to obtain information on the speciation of a given system only at that pH value of the solution.

We also remember that the limiting factor in the study of chemical equilibria is bound to the variation of the activity coefficients as the experimental conditions vary (ionic strength, temperature, etc.). For this reason, the alternative approach used by our and hundreds of papers from the research groups involved in speciation studies is to use the so called “constant ionic medium” that consists in using an ionic medium (strong electrolyte such as NaCl, NaNO3, KCl, NaClO4, etc) at a concentration significantly higher than those of the components in solution. At these conditions, it is assumed that the activity coefficients of the components and of the species have unitary value, allowing to approximate the activities to the concentrations. This allows to investigate the speciation of a given metal-ligand system also if the pH of the solution changes along the titration.

Referee can find more detail in : Chemistry of Marine Water and Sediments. Editors:  Antonio Gianguzza; Ezio Pelizzetti, Silvio Sammartano (2002) - Springer.

Regarding the pL0.5 parameter for which the referee asks for a greater definition, we invite him to read the following manuscript: Crea, F., De Stefano, C., Foti, C., Milea, D. and Sammartano, S. (2014) 'Chelating agents for the sequestration of mercury(II) and monomethyl mercury(II)', Curr. Med. Chem., Vol. 21, No. 33, pp. 3819-3836), where an in-depth discussion is made for this parameter, which is beyond the scope of this manuscript.

  • Also there are many notes about the presentation of the equilibria. The experimental conditions used for this study is far away from the biological conditions and only 0,15 mol/L NaCl and 37 C degree are not enough to modeled them and etc.

It is not clear what the referee means. In any case, we remember that the average ionic strength value of biological fluids is I ~ 0.16 mol dm-3 and the temperature assumed as a reference for this kind of studies is t = 37 ° C (T = 310.15K). Moreover, NaCl was chosen as supporting electrolyte, since it is the main inorganic components of biological fluids and generally used to simulate the interactions that there occur.

For all the three systems studied here, these conditions have been experimentally investigated.

Reviewer 3 Report

The paper "The effect of metal cations on the aqueous behavior of dopamine. Thermodynamic investigation of the binary and ternary interactions with Cd2+, Cu2+ and UO22+ in NaCl at different ionic strengths and temperatures " is a huge comprehensive piece devoted to the very detailed study of the very complex and biologically relevant systems. The authors did just a great work, which is not surprising taking into account their previous experience in the field.

I have just some minor comments regarding this manuscript:

  1. The value -6.90 in Table S3 (at I = 0.75 and T = 298.15 K) is a clear typo, there should be -6.09
  2. The last lines in Table 16 seem to be misplaced.
  3. Most of the trends in Fig. 8 seem to be non-linear. Does it mean the strong dependence of the change in enthalpy on the temperature of is it a consequence of uncertainties of the stability constants?
  4. Talking about the changes in enthalpy. The values of ΔH given in Tables 6, 7 are humongous and comparable with some covalent bonds formation energies. It does not seem correct, especially, when one sees that ΔHM2L = -105.1 kJ/mol and ΔHM2L2 = -267.9 kJ/mol. The addition of the second ligand yields -160 kJ/mol? Really? Another example: ΔHM2L(OH) = 1.6 kJ/mol, which means that the hydrolysis of the M2L complex is endothermic absorbing ~100 kJ/mol. I find these changes in enthalpy doubtful.
  5. Since the constants of side processes are involved in the calculation scheme, their correctness is crucial for the evaluation of the equilibrium constants under study. Sometimes, the chosen side processes' constants deviate from the recommended in the handbook. The most questionable is the choice of constants for UO22+ system as, for example, book by C. Ekberg and P.L. Brown (Hydrolysis of Metal Ions) on P. 351 does not consider the data by De Stefano et al. (2002) realistic regarding (UO2)3(OH)7- formation. Moreover, the paper (10.1021/ic101953q) reports on experimental observation via Raman spectroscopy the set of hydrolyzed species differing from that accounted by the Authors. 
  6. Hydrolysis constants of Cu2+ and Cd2+ also seem to deviate from those reported in the IUPAC technical reports (10.1351/pac200779050895 and 10.1351/PAC-REP-10-08-09). I do not intend to say that the chosen constants are incorrect (it is noteworthy that CdCli constants are in perfect agreement with the tech. report); however, it is necessary to perform all the calculations of the stability constants of metal ions with dopamine using different side reactions constants from different sources and study, how the uncertainty of the side reaction constant influence on the target constants. The error, resulting from the error of the side reaction constants, should be added to that arising from statistical analysis.

The commentary above is not given to diminish somehow the great paper by the Authors; they are caused mostly by my interest in this theme. Some inaccuracies are unavoidable when such a long paper is written. Best of luck to the Authors in developing this field!

Author Response

Reviewer 3

I have just some minor comments regarding this manuscript:

  1. The value -6.90 in Table S3 (at I = 0.75 and T = 298.15 K) is a clear typo, there should be -6.09

Yes. We thank the referee for this indication.

  1. The last lines in Table 16 seem to be misplaced.

The last row of table 16 refers to the formation constants of the Sn2 + / dopamine system. For ease of reading, the authors preferred to write it as reported in this table to avoid adding further columns to the already existing ones.

  1. Most of the trends in Fig. 8 seem to be non-linear. Does it mean the strong dependence of the change in enthalpy on the temperature of is it a consequence of uncertainties of the stability constants?

The authors thank the referee for this careful observation. We believe that the non-linear variation of the formation constants with respect to the temperature at the different ionic strengths, is a concomitance of various factors, probably including the contribution of the enthalpies of formation of the species.

  1. Talking about the changes in enthalpy. The values of ΔH given in Tables 6, 7 are humongous and comparable with some covalent bonds formation energies. It does not seem correct, especially, when one sees that ΔHM2L = -105.1 kJ/mol and ΔHM2L2 = -267.9 kJ/mol. The addition of the second ligand yields -160 kJ/mol? Really? Another example: ΔHM2L(OH) = 1.6 kJ/mol, which means that the hydrolysis of the M2L complex is endothermic absorbing ~100 kJ/mol. I find these changes in enthalpy doubtful.

The authors thank the referee for this observation.

The values of the formation enthalpy of the species seem to be very high, but we remember to Reviewer 3 that they refer to formation enthalpies calculated considering the overall formation constants and not the stepwise ones. Furthermore, these values have been calculated from the knowledge of the formation constants at different temperatures, by means of the van't Hoff term reported in the equation 5.

In order to eliminate any doubts, the authors are planning to carry out the determination of the enthalpies of formation of these systems by means of calorimetric titrations.

  1. Since the constants of side processes are involved in the calculation scheme, their correctness is crucial for the evaluation of the equilibrium constants under study. Sometimes, the chosen side processes' constants deviate from the recommended in the handbook. The most questionable is the choice of constants for UO22+ system as, for example, book by C. Ekberg and P.L. Brown (Hydrolysis of Metal Ions) on P. 351 does not consider the data by De Stefano et al. (2002) realistic regarding (UO2)3(OH)7- formation. Moreover, the paper (10.1021/ic101953q) reports on experimental observation via Raman spectroscopy the set of hydrolyzed species differing from that accounted by the Authors. 

Regarding the speciation model for uranyl ion hydrolysis, the referee made a careful observation.

The study of the behavior of uranyl ion in aqueous solution is particularly complex and has attracted the attention of many researchers for several decades. In all these works, each researcher has proposed very different speciation models, depending on many factors, including pH, ionic medium, metal concentration, etc.

Our research group also studied the hydrolysis of uranyl ion in aqueous solution and in different ionic media (NaCl, NaClO4, NaNO3). The results there obtained are different in term of stability and speciation schemes, since some of the hydrolytic species formed by uranyl are for example stabilized by the chloride ion. This is also confirmed in the volume of the NEA-OECD (CHEMICAL THERMODYNAMICS OF URANIUM) whose main editors are Ingmar GRENTHE, Robert J. LEMIRE, Chinh NGUYEN-TRUNG CREGU, that are researchers of clear world fame in the study of actinides speciation.

The authors understand the possible perplexity of the presence of the species (UO2)3(OH)7-, which Brown does not consider in its book on metal hydrolysis.

However, we want to highlight that this hydrolytic species is also proposed in the volume of the NEA-OECD (CHEMICAL THERMODYNAMIC OF URANIUM), and in particular in table V.7 (selected formation constants for the uranium (VI) hydroxide system); the value there reported at infinite dilution is in agreement with that obtained from our investigation. The existence of this species was also observed by calorimetric titrations and confirmed by other authors by means of RAMAN spectra.

From our investigations, the formation of this species has been only obtained in chloride media and not in perchlorate or nitrate ones.

Concerning the paper proposed by the referee on the hydrolysis of uranium, authors well known this manuscript, where investigation were carried out by Raman and in solution containing a non-complexing electrolyte. The results there obtained are different from those proposed by NEA and in the book of Brown, especially in terms of speciation model.

However, a more accurate treatment of the uranyl hydrolysis was also made in our review: Berto, S., Crea, F., Daniele, P.G., Gianguzza, A., Pettignano, A. and Sammartano, S. (2012) 'Advances in investigation of dioxouranium(VI) complexes of interest for natural fluids', Coord. Chem. Rev., Vol. 256, pp. 63-81.

  1. Hydrolysis constants of Cu2+ and Cd2+ also seem to deviate from those reported in the IUPAC technical reports (10.1351/pac200779050895 and 10.1351/PAC-REP-10-08-09). I do not intend to say that the chosen constants are incorrect (it is noteworthy that CdCli constants are in perfect agreement with the tech. report); however, it is necessary to perform all the calculations of the stability constants of metal ions with dopamine using different side reactions constants from different sources and study, how the uncertainty of the side reaction constant influence on the target constants. The error, resulting from the error of the side reaction constants, should be added to that arising from statistical analysis.

The consideration of the referee is fully correct, regarding the importance of considering, for a correct speciation study, also all the parallel reactions, including the hydrolysis constants and all the eventual species formed with the ions of the ionic medium.

Concerning the referee comment, we want to point out that in 2013 our research group carried out a careful study (review) on the speciation of cadmium in terms of hydrolysis constants and complexes formed with inorganic anions and various classes of organic ligands (Crea F., Foti C., Milea D., Sammartano S. (2013) Speciation of Cadmium in the Environment. In: Sigel A., Sigel H., Sigel R. (eds) Cadmium: From Toxicity to Essentiality. Metal Ions in Life Sciences, vol 11. Springer, Dordrecht. https://doi.org/10.1007/978-94-007-5179-8_3).

This study was carried out considering all the available literature data, including also those reported in the IUPAC report; these data were then re-elaborated.

Concerning the Cu2+ hydrolysis, this were taken from Brown P.L.; Ekberg, C. Hydrolysis of Metal Ions; Wiley: 2016.

However, independently of the considered hydrolysis constants, it is important to observe for the results obtained from our investigations that:

  1. the interaction of dopamine with Cu2+ and Cd2+ begins above pH ~ 4;
  2. below this pH value, copper does not tend to undergo significant hydrolysis, as evidenced by the distribution diagrams;
  3. the Cd2+ is complexed by the chloride ion, inhibiting the hydrolysis of the metal;
  4. throughout the concentration range and molar ratios of the components used, the formation of hydrolytic species is negligible.

Therefore, the consideration that can be made is that, it is certainly correct to consider the hydrolysis constants in the speciation model, but that considering what has been highlighted, they have no influence on the speciation of the investigated metal-dopamine systems.

As a confirmation of our consideration, the potentiometric data of the Cu2+/Dop- system at I = 0.15 mol dm-3 and T = 298.15 K were re-processed by using the following hydrolytic constants for Cu2+:

I/mol L-1

logbCuOH

logbCu(OH)2

logbCu(OH)3

logbCu(OH)4

logbCu2(OH)2

0.15

-8.242

-17.555

-27.806

-39.053

-10.633

These data were taken from: Baes, C. F.; Mesmer, R. E., The Hydrolysis of Cations. John Wyley & Sons, New York: 1976

The result obtained at I = 0.15 mol dm-3 and T = 298.15 K for the Cu2+/Dop- system by using this “new” hydrolytic species is:

I/mol dm-3

logbML2 

logbM2L

logbM2L2

logbM2L2(OH)2 

logbM2L(OH)

logbML2(OH)

T = 298.15 K

0.171

19.39±0.07

14.50±0.04

25.72±0.08

11.76±0.05

8.91±0.06

10.95±0.08

Within the experimental error (Std. Dev.), the “new” values are perfectly in agreement with those reported in the manuscript.

The results here reported confirm what above reported, namely, it is important to consider for a correct speciation study, also the parallel reactions, but that as in this case, the strength of the interaction between the metal and dopamine, avoid the hydrolysis of the metal, with a result that they do not influence the speciation of the metal/ligand systems and the stability constant result fairly unchanged.

The commentary above is not given to diminish somehow the great paper by the Authors; they are caused mostly by my interest in this theme. Some inaccuracies are unavoidable when such a long paper is written. Best of luck to the Authors in developing this field!

We want to thank the referee for its constructive comments.

Round 2

Reviewer 1 Report

The authors give appropriagte answers for the questions, but the speciation of the systems is not suficiently supported by spectroscopic measurements. However, taking into account that thermodynamic data are only scarcely available for the cadmium(II) and uranyl complexes, I may recommend publication.

Reviewer 2 Report

After carefully reading the revised version of the manuscript ID: molecules-1485928, as well as the author's answers, I cannot support its publication.  It seems that the processing is only superficial. The authors put more effort into arguing with the reviewer instead of improving the work in terms of research, presentation of results and relevance.

I could not support a scientific approach based on the same study, in which the cations are changed once (Biomolecules 2021, 11, 1312.) and then the organic ligands are changed (Molecules 2019, 24, 4084), and the research system is artificial and the results are inapplicable in practice.

I would like to reiterate my advice to the authors, namely to include in the team a specialist in the field of analytical and coordination chemistry, as there are many incorrect and inaccurate statements from the classical chemistry point of view.